# Clinicopathological Pearls and Diagnostic Pitfalls in IgG4-Related Disease: Challenging Case Series and Literature Review

**DOI:** 10.3390/diagnostics15182299

**Published:** 2025-09-10

**Authors:** Sokol Sina, Giulio Luigi Bonisoli, Sofia Vitale, Luigi Marzano, Stefano Francesco Crinò, Maria Cristina Conti Bellocchi, Sara Boninsegna, Simone Conci, Federica Maiolini, Riccardo Nocini, Luca Sacchetto, Giorgio Barbera, Andrea Fior, Nikela Kalaja, Elena Malloggi, Antonietta Brighenti, Alice Parisi, Nicolò Cardobi, Aldo Scarpa, Simonetta Friso, Elisa Tinazzi

**Affiliations:** 1Section of Pathology, Department of Diagnostics and Public Health, University of Verona, Piazzale L.A. Scuro 10, 37134 Verona, Italy; antonietta.brighenti@aovr.veneto.it (A.B.); alice.parisi@aovr.veneto.it (A.P.); aldo.scarpa@univr.it (A.S.); 2Section of Internal Medicine B, Department of Medicine, University of Verona, Piazzale L.A. Scuro 10, 37134 Verona, Italy; giulioluigi.bonisoli@univr.it (G.L.B.); sofia.vitale@studenti.univr.it (S.V.); luigi.marzano@aovr.veneto.it (L.M.); federica.maiolini@aovr.veneto.it (F.M.); nikela.kalaja@aovr.veneto.it (N.K.); elena.malloggi@studenti.univr.it (E.M.); simonetta.friso@univr.it (S.F.); elisa.tinazzi@univr.it (E.T.); 3Diagnostic and Interventional Endoscopy of Pancreas, The Pancreas Institute, University Hospital of Verona, 37134 Verona, Italy; stefanofrancesco.crino@aovr.veneto.it (S.F.C.); mariacristina.contibellocchi@aovr.veneto.it (M.C.C.B.); 4Gastroenterology Unit, IRCCS Sacro Cuore Don Calabria Hospital, 37024 Negrar, Italy; sara.boninsegna@sacrocuore.it; 5General and Hepatobiliary Surgery, Department of Surgey, Dentistry, Gynecology and Pediatrics, University of Verona, 37134 Verona, Italy; simone.conci@univr.it; 6Unit of Otoryngology-Head and Neck Departement, University of Verona, Piazzale L.A. Scuro 10, 37134 Verona, Italy; riccardo.nocini@aovr.veneto.it (R.N.); luca.sacchetto@univr.it (L.S.); 7Maxillo Facial Surgery Unit, Head and Neck Department, Azienda Ospedaliera Universitaria Integrata Verona, 37134 Verona, Italy; giorgio.barbera@aovr.veneto.it (G.B.); andrea.fior@aovr.veneto.it (A.F.); 8Department of Radiology, G.B. Rossi Hospital, University of Verona, 37134 Verona, Italy; nicolo.cardobbi@univr.it

**Keywords:** IgG4-related disease, immunoglobulin G4, immune-mediated fibroinflammatory disease, autoimmune pancreatitis, IgG4-related sclerosing cholangitis, retroperitoneal fibrosis and related disorders, salivary gland disease, Dacryoadenitis, ocular and orbital inflammatory disease, lymphadenopathy, Atypical features of IgG4-RD

## Abstract

**Background:** IgG4-related disease (IgG4-RD) is a chronic immune-mediated fibroinflammatory disorder characterized by lymphoplasmacytic infiltrates enriched in IgG4-positive plasma cells, storiform fibrosis, and frequently elevated serum IgG4 levels. Classic forms, such as pancreaticobiliary or retroperitoneal involvement, are often recognized early, whereas atypical manifestations mimic malignancy or inflammatory conditions, leading to delayed or inappropriate treatment. **Case Series**: A 30-year-old man presented with hyperemesis, proptosis, and gait instability. He was found to have colonic stenosis, stomach thickening, pachymeningitis, and polyserositis. Gastroenteric histology and serology confirmed IgG4-RD. Steroids were ineffective, but rituximab produced sustained clinical and radiologic improvement. A 35-year-old woman developed jaundice and cholestasis with a perihilar mass highly suggestive of cholangiocarcinoma. Histopathology revealed IgG4-RD, and rituximab therapy led to marked clinical and serological improvement. A 64-year-old woman with a submandibular mass underwent sialoadenectomy, with histology confirming IgG4-RD; she remained asymptomatic without systemic treatment. **Literature Review**: A literature review highlighted the diagnostic challenges of atypical IgG4-RD. Gastrointestinal involvement is rare and often misclassified as inflammatory bowel disease. Isolated biliary disease frequently mimics cholangiocarcinoma, while salivary gland involvement may be misdiagnosed as neoplasia. Serum IgG4 levels >135 mg/dL and IgG4/IgG ratio >0.21 may support clinical suspicion, but histopathology remains indispensable for definitive diagnosis and for excluding malignancy. Steroid responsiveness is a hallmark, though relapses after tapering are common, often necessitating B-cell-directed therapy. **Conclusions**: IgG4-RD should be considered in patients with unexplained, relapsing, or steroid-responsive conditions. Early recognition, multidisciplinary collaboration, and integration of histopathology with clinical features are essential to avoid misdiagnosis and optimize management.

## 1. Introduction

IgG4-related disease (IgG4-RD) is a rare and probably underrecognized (estimated incidence of 1.4/100,000 person and prevalence of 5.3/100,000 persons [1]) immune-mediated disorder characterized by fibrosis and potentially affecting nearly any organ or system. The fibrotic process is associated with an IgG4-positive lymphoplasmacytic infiltration within the tissue and is often associated with serum IgG4 levels > 135 mg/dL [2,3].

The disease may present either with single-organ involvement or as a multisystem condition, affecting three or more organs simultaneously or sequentially [4]. Pancreas, biliary ducts, retroperitoneum, kidneys, salivary and lacrimal glands, orbits, thyroid, lungs, aorta, and serous membranes are the organs most commonly involved [2,5]. Multisystem IgG4-RD occurs in up to 90% of patients and typically manifests in one of four distinct phenotypes [4]:pancreato-biliary disease, 31% of the patients;retroperitoneal-aortitis, 24% of the patients;head and neck disease without other features, 24% of the patients;head and neck disease with extra-glandular involvement, 22% of the patients.

While specific clinical phenotypes, such as pancreatic or retroperitoneal involvement, may help guide the classification, the frequent presence of heterogeneous features or rare organ involvement often leads to missing the diagnosis or the need for additional evidence. As highlighted in several studies, the disease may affect organs beyond the most commonly involved ones, with numerous atypical forms closely mimicking other conditions [6,7,8,9]. Furthermore, fibrosis can be so extensive as to generate mass-forming lesion, frequently leading to an initial misdiagnosis of malignancy or other diseases [10].

Laboratory findings, although sensitive, often lack specificity. Elevated serum IgG4 levels are commonly observed not only in patients with IgG4-RD, but also in individuals with atopy, other autoimmune diseases, infections, and malignancy [11,12,13,14]. Additionally, hypereosinophilia, usually lower than 1000 cell/µL, may be present, imposing differential diagnosis with hypereosinophilic syndromes [15]. Peripheral blood plasmablast count is also useful for the diagnosis and potentially related to diagnostic suspicion [16,17].

Given the variability of clinical phenotypes and the unreliability of laboratory markers, histopathology is essential to confirm the diagnosis, leading to the ruling out of malignancy and verification of clinical hypothesis [18]. The histopathological hallmarks of the disease are fibrosis, lymphoplasmacytic infiltrate, and obliterative phlebitis [11,19]. Fibrosis is considered a prerequisite for diagnosis and is almost invariably present, either extensively or in patches [19]. The typical pattern is storiform fibrosis; however, since organ involvement is often patchy, this pattern may be absent in smaller biopsy samples [11,19]. The classic lymphoplasmacytic infiltrate is defined by more than 10 IgG4-positive plasma cells per high-power field (HPF—corresponding to a magnification of 40×) and by an IgG4/IgG ratio greater than 0.4 [11,19]; moreover, the differential diagnosis with other diseases associate with IgG4-rich infiltrates is to be considered [11,19,20]. Plasma cells are often associated with eosinophilic infiltration, although the eosinophil levels are characteristically lower than those observed in hypereosinophilic syndromes [11,15,19]. Lastly, obliterative phlebitis, usually sparing arteries, is characterized by venous infiltration without necrotizing vasculitis [11,19].

Finally, it is important to underline the absence of diagnostic criteria with acceptable sensitivity and the common use of classification criteria for diagnosis. The 2019 EULAR classification criteria are useful to assess the patient’s inclusion in clinical studies, meaning they are intended for research purposes only, not for diagnosis [2]. As EULAR criteria do not take into account the rarer clinical features of the disease, as outlined below, they are characterized by a relatively low sensitivity (82%) and therefore are inadequate for diagnostic purposes [1,19]. As reported by EULAR commission, the non-adherence to the classification criteria should not rule out the diagnosis if it is clinically appropriated [2].

We present a case series of three patients with a very unusual clinical presentation of IgG4-RD that did not fit into the current classification criteria and were solved only after multidisciplinary discussion between clinician and pathologist. This offers an insight into the complexity of this disease, that is often misdiagnosed as other nosological entities. We also performed a review of the literature to highlight the most unusual and overlooked clinical features of the disease to aid the diagnostic process.

## 2. Case Presentation

The main features of the three cases are summarized in Table 1. Pictures related to the cases can be found in the Appendix A.

### 2.1. Case 1

#### 2.1.1. Clinical Presentation

A 30-year-old man presented with progressive hyperemesis, which had worsened through a period of two months. He had also experienced a progressive proptosis with blurred vision and gait instability. The patient worked as a delivery boy, had no significant risk factors, since he denied smoking and alcohol consumption, and denied foreign travel except for one trip to Sri-Lanka in the previous year. His only notable medical history was a surgically drained subdural traumatic hematoma two years before hyperemesis onset.

Hyperemesis was accompanied by constipation, abdominal distension, coffee-ground vomiting, and being unresponsive to any antiemetic therapies. Blood tests revealed elevated inflammatory markers, mild eosinophilia, and sightly increased IgG4 levels (334 mg/dL) without any other significant findings.

#### 2.1.2. Investigation

Ultrasonography detected ascites that prompted further investigation by abdominal CT-scan (Appendix A) and MRI, which confirmed ascites associated with pleural effusion, inflammatory sub-stenosis of the sigmoid colon, and patchy ileal thickening. PET-CT demonstrated diffuse intestinal uptake. With an initial clinical suspicion of Chron disease, the patient underwent esophagogastroduodenoscopy (EGDS—Appendix A) and colonoscopy that were both inconclusive except for the finding of inflammatory mucosal alteration with hyperemia and lymphoplasmacytic and eosinophilic infiltrates at the biopsy specimen evaluation of both sites.

Progressive vomiting worsening led to a brain MRI that revealed a dural thickening suggestive for pachymeningitis accompanied by residual signs of the previous hemorrhage, and surgical intervention. Cerebrospinal fluid (CSF) analysis indicated inflammation without evidence of infection. These findings shifted the initial clinical suspicion to that of a possible eosinophilic vasculitis; prednisone (1 mg/kg/day) was, therefore, initiated. One month later, a clinical improvement was noticed, although during steroid tapering, symptoms relapsed, while imaging was persistently consistent with the presence of sigmoid stenosis. Exploratory laparoscopy revealed reduced bowel mobility and a 10 cm omental mass, which was excised. Histology demonstrated fibrosclerosis and inflammatory microangiopathy with mixed eosinophilic and plasma cell infiltrates, with no significant positivity of plasma cells to IgG4 (<10 high-power fields—HPF) (Figure 1).

#### 2.1.3. Histopathology

As symptoms persisted, the patient was transferred to our Internal Medicine Unit for further evaluation. A total body CT-scan confirmed polyserositis (pachymeningitis, ascites and pleural effusion). Bone marrow biopsy (BMB), ascitic fluid analysis, and PET-CT excluded hematological malignancies. Blood tests showed elevated serum IgG4 levels (366 mg/dL), without eosinophilia or clear evidence of infectious disease. A further gastric biopsy revealed signs of chronic inactive gastritis, with minimal eosinophilic infiltrate. Multidisciplinary discussion among internists and pathologists that considered either histology or clinical features, prompted the reconsideration of IgG4-RD. Immunohistochemistry revealed a significant IgG4 positive plasma cell infiltrate (>10 cell/HPF) with an IgG4/IgG ratio > 0.4; furthermore, a wall-thick biopsy ruled out eosinophilic vasculitis.

#### 2.1.4. Treatment

Based on high IgG4 blood levels, polyserositis, histological findings, and the exclusion of malignant hematological disease, a diagnosis of atypical IgG4-RD was made. Considering the preceding steroid therapy failure, rituximab treatment (1 gram with a repeat dose after 2 weeks) was initiated, resulting in symptomatic improvement within ten days.

#### 2.1.5. Outcome

After the second rituximab dose, all symptoms resolved.

After six months, blurred vision persisted with worsening of pachymeningitis on MRI, although no signs of intestinal involvement were evident. Serum IgG4 levels increased to 450 mg/dL, prompting indication to a second rituximab cycle with same protocol as previous administration, that led again to clinical improvement, with a decrease in serum IgG4 levels to 345 mg/dL associated with an improvement in meningitis signs at MRI. Follow-up endoscopic evaluation revealed normal gastric mucosa with histological resolution of the previously described infiltrate.

### 2.2. Case 2

A 35-year-old woman came to our hospital due to asthenia, loss of appetite, generalized pruritus, jaundice, and colorless stools accompanied by dark urine. Blood tests revealed anemia along with elevated markers of hepatic cytolysis and cholestasis. An abdomen CT-scan revealed a 3.5 cm mass-forming lesion surrounding the main bile duct, infiltrating the hepatic artery and portal vein, with associated splenomegaly and two satellite lesions in the gallbladder and right lung, all findings highly suspicious for perihilar cholangiocarcinoma. An endoscopic ultrasound (EUS) and endoscopic retrograde cholangiopancreatography (ERCP) confirmed the presence of an ill-defined 3.5 cm mass causing a type II hilar bile ducts narrowing, near to the biliar carrefour (Appendix A). A biliary stent was placed, and a fine needle biopsy (FNB) was performed. Histological examination ruled out cholangiocarcinoma, revealing only nonspecific fibroconnective tissue. Given those findings, serum IgG4 levels were measured and found to be slightly elevated (135 mg/dL).

Subsequently, the patient underwent exploratory laparoscopy, which revealed two solid lesions in the spleen and a firm, infiltrative lesion at the hepatoduodenal ligament involving vascular structures. Intraoperative histological evaluation of the gathered specimen biopsies was inconsistent with initial suspicion of an epithelial neoplasm. Histology revealed instead storiform fibrosis, vascular inflammation, and a rich plasma cell infiltrate that was IgG4 positive (>10 cell/HPF, ratio > 0.4) (Figure 2). One month later serum IgG4 levels remained stable at 150 mg/dL. The patient was then transferred to our Internal Medicine Unit with a histological diagnosis of IgG4-RD. Methylprednisolone at the dosage of 1 g intravenous pulses were administered for three consecutive days, and subsequently rituximab—1 g infusion with repetition after 2 weeks—was initiated as a steroid-sparing agent. Five months after the beginning of the therapy, an abdominal CT-scan showed a reduction in lesion size, serum IgG4 levels dropped to 74 mg/dL, and the clinical status of the patient had normalized.

### 2.3. Case 3

A 64-year-old woman reported a progressively enlarging submandibular mass for approximately one year, that was suspicious for neoplastic lesion. The patient had no significant risk factors except for a long time lasting essential arterial hypertension. Ultrasonography revealed that the mass pertained to the submandibular gland and grew over time to a diameter of 2.5 cm and was associated with satellite enlarged lymph nodes (Appendix A). A fine needle aspiration excluded the presence of neoplastic cells. Nevertheless, due to the persistent suspect of a possible malignant nature of the lesion, the patient underwent a sialoadenectomy. The histopathological examination demonstrated extensive fibrosis with glandular atrophy, obliterative phlebitis, and plasma cell infiltrate rich in IgG4-positive plasma cell (countless number of cells per HPF, with an IgG4/IgG ratio > 0.4) (Figure 3). Postoperative serum IgG4 levels were within the normal range (74 mg/dL); unfortunately, the preoperative measurement was not available. PET-CT-scan showed hypermetabolism at the right colon that was suggested to be inflammatory, with no other hypermetabolic site. Colonoscopy histology revealed mild unspecific chronic colitis. Based on these findings a diagnosis of localized IgG4-RD was established, the patient was managed with follow-up without needing systemic therapy. In the following months, no other signs of disease appeared.

## 3. Case Series Discussion and Literature Review

IgG4-RD frequently involves the pancreas, retroperitoneum, bile ducts and salivary glands [21,22,23]. Its most characteristic—and therefore extensively studied—feature is type 1 autoimmune pancreatitis (AIP1), which is readily suspected in both clinical practice and pathological examinations [24]. AIP1 is a rare disease with a slightly higher prevalence among middle-aged and elderly men [24,25]. According to Gallo et al. it has an insidious onset, with obstructive jaundice and other nonspecific symptoms, such as diarrhea and weight loss [26]. Pancreatic involvement can be so severe that it compromises endocrine function, ultimately leading to onset of diabetes mellitus [24,26].

This work aims to highlight some of the most overlooked manifestations of the disease, focusing on characteristics that may either mislead or guide the clinicians toward an accurate and timely diagnosis. Table 2 summarizes the main characteristic for each organ involvement considered.

It must be highlighted that each organ involvement presents a different threshold to be considered as a feature of IgG4-RD, due to the usual histological characteristic of the disease and to avoid misdiagnosis of other histologically similar entities. All reported values should be considered indicative and were extrapolated from the currently available scientific literature.

### 3.1. Case 1

#### 3.1.1. IgG4 Related Gastroenteric Disease (GE-IgG4-RD)

GE-IgG4-RD is among the rarest features of the IgG4-RD, and its classification still remains controversial [14,66,67]. A systematic review by Sawada et al. summarized the main findings in 42 cases of GE-IgG4-RD up to 2022 [68]; however, overall, the literature is based on single case reports or case series [62,69,70,71,72].

GE-IgG4-RD is often paucisymptomatic and usually diagnosed incidentally [62,73]. Patients may complain of abdominal pain and asthenia, and sometimes present with vomit and diarrhea or constipation [62,68,70]. The initial clinical suspicion is typically Chron’s disease or malignancy, potentially leading to inappropriate surgical intervention, often before serum IgG4 levels are obtained [61,68,72]. As described, Case 1 patient fits the presentation with severe vomiting and alteration in the bowel habits.

As in the case of the Case 1 patient, laboratory findings are usually nonspecific, with only a mild increase in CRP and ESR, commonly searched autoantibodies—such as ANA, anti-ENA, and anti-transglutaminase—are usually result negative [61]. Fecal calprotectin may be increased [61]. Serum IgG4 levels are typically found to be increased (>135 mg/dL), and it also emerged that serum IgG4/IgG ratio >25% can be helpful to diagnostic path for the clinician suspicion [46,68]. However, it should be noted that other diseases can be associated with an increase in serum IgG4 concentrations, and some IgG4-RD-affected patients may, as a counterpart, exhibit only a mild IgG4 increases [46,61]. The Case 1 patient had a severe increase in serum IgG4 level (>300 mg/dL), which prompted, after discarding IBD hypothesis, the search of a GE-IgG4.

CT-scan may reveal mass-forming lesions, frequently misdiagnosed as a gastrointestinal stromal tumor (GIST) [62,68]. Focal or diffuse gastroenteric wall thickening—which can cause sub-stenosis—and presence of gastric or bowel nodules have also been described [61,68]. Patient of Case 1 did not present any focal mass-forming lesion of the bowel but had stricture with bowel and gastric wall thickening. Ruling out solid tumors and lymphomas is essential [46,68]. PET-CT can demonstrate diffuse tracer uptake corresponding to active disease sites, although misinterpretation of focal enhancement may erroneously suggest a diagnosis of neoplasm or IBD [74,75,76].

Endoscopy is critical for assessing mucosal status and obtaining biopsy specimen to exclude other gastroenteric diseases such as IBD. Sawada et al. reported that submucosal mass is the most common finding, whereas Yoshidome et al. also observed chronic mucosal inflammation, often accompanied by ulceration or polypoid lesions [68,72], making IBD the main differential diagnosis, as in the case of the Case 1 patient.

The key histopathological hallmark of GE-IgG4-RD is bottom-heavy lymphoplasmacytic infiltration, but it is present only in a fraction of patients (29–42%) [68]. The main diagnostic criteria include at least 10–50 IgG4 positive plasma cell per HPF or an IgG4/IgG ratio exceeding 0.4–0.5 [30,61,68].

There are currently no conclusive diagnostic criteria for GE-IgG4-RD, and the presence of IgG4 positive plasma cells is also common in other inflammatory bowel conditions [30,67]. Herein, when GE-IgG4-RD is suspected, the clinician should actively search for other organ involvement both from a clinical and a pathological point of view [62,72]. Moreover, according to the HISORT criteria [77], a positive response to steroid therapy, despite potential relapses at withdrawal, can be considered supportive evidence for the diagnosis [61,67,72].

We hypothesize that, in patients with suspected GE-IgG4-RD, a histological threshold of 10 IgG4 positive plasma cell per HPF, combined with an IgG4/IgG ratio greater than 0.4, elevated serum IgG4 levels and additional clinical features—such as in the Case 1 patient who also had pachymeningitis, polyserositis, and ocular disease—moreover when combined with a prior response to steroid therapy, may provide sufficiently specific evidence to support the diagnosis, even in the absence of a definitive pathological confirmation from other organs. In this setting, given the rarity of this type of organ involvement, it is crucial to discuss with the pathologist and the gastroenterologist to agree on the search of IgG4 plasma cells in biopsy and avoid the pitfalls of IBD-like presentation of the disease.

#### 3.1.2. IgG4-Related Sclerosing Mesenteritis (SM-IgG4-RD)

Sclerosing mesenteritis is a very rare fibroinflammatory condition characterized by necrosis and fibrosis which can lead to focal or diffuse thickening, or mass-forming lesions of the mesenteric adipose tissue [53,54,55]. Up to half of the cases SM may be attributable to an underlying IgG4-RD (SM-IgG4-RD), and are often accompanied by involvement of other organs [54,78]. In fact, isolated SM-IgGRD is even rarer, and its description in the literature remains largely anecdotical [53,79,80].

SM-IgG4-RD is often detected incidentally during evaluation for other conditions or as part or staging studies for IgG4-RD with multi-organ involvement. Clinical findings are usually nonspecific, including weight loss, diarrhea or constipation, abdominal pain, and loss of appetite [54,56,80], symptoms shared by GE-IgG4-RD and that the Case 1 patient also complained. If left untreated, the disease can progress to cause ascites and bowel obstruction [57,58], which we suspect may explain the ascites of the Case 1 patient. Laboratory findings are nonspecific, revealing only a mild increase in inflammatory markers, along with an elevation in serum IgG4 levels, usually exceeding 135 mg/dL [53,54,56].

US is usually nonspecific, even though it may reveal well-defined masses [54]. CT-scan usually shows increased density of the mesenteric fat tissue and presence of well-defined mass-forming lesions without an infiltrative appearance [54,80,81]. MRI can detect masses on T1-weighted images and demonstrate fibrosis or edema as hypointense or hyperintense areas on T2 weighted images [54,81]. ^18^FDG-PET-CT may be negative or reveal increased uptake areas corresponding to the mass-forming lesions, often raising suspicion for malignancy [56,81].

Differential diagnosis includes neoplastic lesions, as well as edema, lymphedema, and hemorrhages. Lymphadenopathy, particularly lymph nodes of >10 mm in diameter or splenomegaly should raise the diagnostic suspicion of an underlying lymphoma [54,78]. A detailed clinical history is therefore crucial in differentiating SM-IgG4-RD from other diseases, excluding causes of edema or hemorrhage.

Histological evaluation is essential and should not be delayed when the clinical suspicion is strong [54]. Histological analysis usually shows a chronic inflammatory infiltrate with fibrosis exhibiting a patchy storiform pattern and fat necrosis; often, there is the evidence of obliterative phlebitis [54,55]. Immunochemistry reveals IgG4-positive plasma cells infiltrate with more than 10 cells per HPF and an IgG4/IgG ratio exceeding 0.4 [53,79].

None of the imaging revealed the mass-forming-lesion of mesentery of the Case 1 patient, which was detected in laparoschopy. Furthermore, at the time of the mass histological analysis, the patient was undergoing steroid therapy, which could have impaired the results. Therefore, considering all the clinical aspects and the outcomes, we suspect that mesenterial involvement was present in the Case 1 patient. The discussion with the pathologist, considering the exclusion of other diseases, A multidisciplinary discussion with the pathologist, given the exclusion of alternative diagnoses, might have redirected the diagnostic framework toward the correct clinical suspicion, despite steroid therapy having already been initiated.

#### 3.1.3. IgG4-Related Orbital Disease (O-IgG4-RD)

Orbital tissues are frequently involved in IgG4-RD, affecting approximately 23% of the patients [40,82]. Studies estimate that up to 50% of cases previously diagnosed as idiopathic orbital inflammation may be an unrecognized O-IgG4-RD [41,83]. In a literature review by Wu et al., more than half of the patients presented with bilateral involvement, finding also a correlation with development of a systemic disease [40]. The most reported symptoms are painless swelling of eyelids, proptosis, diplopia, and restricted eye movements [40].

Proptosis, diminished visual acuity, and ocular movement limitation are often due to orbital tissues swelling, secondary to infiltration and inflammation [41,42]. Commonly involved retroorbital structures include orbital fat tissue (29–40%), extraocular muscle (19–25%) and trigeminal nerve (10–39%), particularly the infraorbital branch [40,41,43,44,84]. The Case 1 patient reported a progressive swelling of eyes, associated with diminished visual acuity and sometimes diplopia, which might in fact represent an orbital involvement do the disease.

CT findings are typically nonspecific, ranging from irregular infiltration to ill-defined lesions or clear tumor-like masses in the orbital fat [43,44,45]. Extraocular muscles (EOMs) can also be affected, appearing asymmetrically enlarged, mimicking a pseudotumor or forming tube-like lesions [42,45,85]. Cranial nerve may be involved with minimal or no neurological deficits, with imaging findings similar to those of muscles, including infiltration or mass-forming lesions [41,83]. No direct orbital imaging was taken during the investigation of the Case 1 patient.

Pathological data in the retro-orbital disease is mostly derived from single case reports. The most accepted diagnostic criteria for O-IgG4-RD include the evidence of fibrosis, lymphoplasmacytic infiltrate, and IgG4/IgG ratio exceeding 40% [43,86].

Dacryoadenitis is the most common orbital manifestation, occurring in 60–90% of patients O-IgG4-RD. It is often bilateral and associated with salivary gland involvement [41,42]. IgG4-RD dacryoadenitis is typically ANA- and anti-Ro52 negative, with xeropthalmia less frequently reported [41]. On CT, it appears as symmetrical, diffuse glandular enlargement with homogeneous density and contrast enhancement, although masses or nodules may also be seen [85]. Pathological criteria for diagnosis include evidence of fibrosis and more than 40–100 plasma cell per HPF, or an IgG4/IgG ratio exceeding 40% [43,86,87].

#### 3.1.4. IgG4 Related Neurological Involvement

Pachymeningitis is a rare complication of IgG4-RD, affecting approximately 2% of patients, yet it accounts for a significant part of cases previously diagnosed as idiopathic hypertrophic pachymeningitis [82]. Dural involvement may be the sole manifestation of the disease or occur as part of a multisystemic process; indeed, in about 40% of cranial IgG4-RD, meninges are also affected [47,48]. Symptoms vary, depending on the site of meningeal involvement [48,49]. Patients may be asymptomatic or present with signs related to vascular or nerve compression, or with symptoms of intracranial hypertension—such as headache, asthenia and ataxia, seizure, and cognitive impairment [47,48,49].

Brain CT-scan may reveal localized or diffuse thickening of the dura mater, or even a meningioma-like mass-forming lesion [47]. MRI can identify T2-hypointense thickening of dura mater, associated hyperintense foci, corresponding to inflammatory sites; often it also shows gadolinium-enhanced T1 lesions mimicking meningiomas [44,47,49]. In this context other possibilities, such as lymphoid or other malignant disease must be excluded.

Histopathologically, pachymeningitis in IgG4-RD is characterized by fibrosis, accompanied with an IgG4 positive lymphoplasmocytic infiltrate exceeding 10 cell/HPF an IgG4/IgG ratio greater than 40% [20,47,48]. CSF analysis may reveal a mild protein elevation and lymphocytosis [49], while oligoclonal IgG4 bands may be present in the active phase of the disease, disappearing at remission [47,50].

The Case 1 patient had both ataxia and asthenia; imaging was coherent with a pachymeningitis that ameliorated with IgG4-RD therapy. Therefore, we believe he might have had pachymeningeal involvement.

True central nervous system involvement is rather rare and, predominantly, results from the extension of pachymeningitis or pituitary involvement. The latter typically manifests with hypopituitarism, diabetes insipidus, and hyperprolactinemia [47]. Brain MRI may highlight pituitary thickening of the hypophysis with T1 and T2 hypointensity and gadolinium enhancement [82].

### 3.2. Case 2

#### IgG4 Related Sclerosing Cholangitis (SC-IgG4-RD)

SC-IgG4-RD is the primary biliary manifestation of IgG4-RD and the second most overall common feature [4,21,88]. It typically affects men around the age of 60 (range 23–88 years) [89]. Symptoms are primarily linked to strictures of bile ducts, obstructing bile flow, causing jaundice, upper abdominal pain episodes, and weight loss [21,88]. Depending on the bile ducts stenosis site it is classified in five types [34,90]:Type 1, intrapancreatic disease.Type 2, diffuse extrahepatic and intrahepatic disease, with (type 2a) or without (type 2b) pre-stenotic dilatation of bile duct.Type 3, hilar stenosis associated with distal stenosis.Type 4, hilar only stenosis.

The association of Type 1 SC-IgG4-RD with AIP1 is debated. The alteration in biliary ducts might result from direct compression by inflamed pancreas tissues (AIP1—83–92%), rather than an intrinsic bile duct disease [21,88,91]. Naitoh et al. recently observed a significantly greater plasma cell infiltration in the bile ducts of patients with non-AIP1-associated SC-IgG4-RD, which also appears to be more prevalent in women [92].

Due to the overlapping of clinical characteristics with other biliary diseases, thorough differential diagnosis is essential. It is a critical issue to distinguish SC-IgG4-RD types 1, 3 and 4 from malignancy (cholangiocarcinoma—CCa) and rule out primitive sclerosing cholangitis (PSC) [93,94,95,96]. For instance, the first clinical suspicion in the Case 2 patient was CCa, and she had to undergo both EUS with FNAB and laparoscopy to exclude it.

An elevated serum IgG4 level (>135 mg/dL) can support the diagnosis, as it is typically higher in SC-IgG4-RD patients [14,35]. However, diagnosis cannot rely only on IgG4 levels, as elevation in serum IgG4 concentration may occur in both CCa and PSC [13,14]. Specificity increases with serum concentration, peaking it gets to four times the normal limit, although at the cost of sensitivity [13,14]. Boonstra et al. proposed that IgG4/IgG1 ratio exceeding 0.24 might help to differentiate SC-IgG4-RD from PSC [97]. On the other hand, relatively low serum IgG4 levels, as highlighted by Case 2 patient, do not exclude the disease, especially in patients who already are on steroid therapy, as the negative predictive value is insufficient [98,99,100,101]. Vosskuhl et al. suggest that bile IgG4 levels over 13 mg/dL may distinguish SC-IgG4-RD from PSC and CCa [102].

Low, nonspecific ANA titer (1:40–1:80) may be present in SC-IgG4-RD [103]; however, higher titers are linked with other autoimmune cholestatic diseases, such as primary biliary cholangitis—which is linked to multiple nuclear dots and rim-like/membranous ANA patterns—or PSC [104]. It is also worth mentioning that ANA may present in many other diseases, such as infection by HBV and HCV [105,106]. Therefore, ANA presence should prompt other investigations to identify different diseases.

Strictures of the bile ducts and mass-forming lesions are the typical imaging findings of SC-IgG4-RD [21,107]. On CT, diffuse or long-segmental strictures, frequently exhibiting a “funnel-shaped” appearance and concentric thickening of bile duct wall, are more commonly observed in SC-IgG4-RD than in CCa [21,34,35]. Experienced radiologist may differentiate SC-IgG4-RD, which exhibit a single-layered contrast enhancement, from CCa, which shows double layered enhancement [35]. CCa instead usually is characterized by a focal, ill-defined mass or nodular thickening, with non-widespread bile duct branch dilation, associated with vascular invasion [108].

MRI findings in SC-IgG4-RD usually show a thickening of bile ducts wall [109]. In fact, a common bile duct thickness > 2.5 mm may be suggestive of SC-IgG4-RD in contrast enhanced imaging [109]. In patients with more than one tract involved by the disease, the interposed segments shows an abnormal thickening [109]. Conversely PSC is characterized more by intrahepatic skip lesions with tracts of bile ducts that seem to not be involved in the disease [109].

In PSC, MRCP or ERCP may reveal band-like structures, a beaded appearance, or pruned tree pattern, alternating normal or slightly dilated segments [108,110]. In contrast SC-IgG4-RD, usually features funnel-shaped, long-segmental stenosis [21,34,111]. EUS and intraductal ultrasound (IDUS) are useful tools to evaluate bile duct walls. The hallmark of SC-IgG4-RD is an abnormal symmetric and smooth increase in bile duct wall thickness that extend to non-stenotic sections [108,109]. Moreover, both EUS and IDUS enable the operator to perform targeted biopsies, as observed in the Case 2 patient, aiding in the exclusion of other diseases and raising the suspicion for IgG4RSC [112,113,114,115].

Histopathological examination reveals storiform fibrosis, with lymphoplasmacytic infiltration, often associated with eosinophils. Immunohistochemistry reveals IgG4 positive plasma cells infiltrate (>10 plasma cells/HPF in biopsies and >50 cells/HPF in surgical specimens), with an IgG4/IgG ratio > 0.4 [20,34].

### 3.3. Case 3

#### IgG4 Related Sialadenitis (S-IgG4-RD)

Salivary glands are among the most common site affected by the disease, being involved in up to half of the patients [36,37]. Most of the patients present with bilateral mass-forming lesions, whereas unilateral lesions, often mistaken for tumors as in the case of the Case 3 patient, are less frequent [36,37,38]. Other head–neck structures may be involved, either synchronously or metachronously [38,116]. Even though glandular involvement can be asymptomatic aside from swelling, more than one-third of the affected patients experience xerostomia [36,37].

Patients with glandular disease may also have extra-salivary involvement, such as lungs or lymph nodes [36]. The main differential diagnoses include Sjogren syndrome, lymphomas, and other head–neck malignancy [36]. S-IgG4-RD patients typically have negative or low autoantibodies titers [37,38], while high ANA or anti-ENA titers, especially with SSA or SSB specificity, suggest alternative diagnoses [2,39].

Imaging (CT, MRI and US) in most patients reveals homogeneous bilateral enlargement of the submandibular, parotid, or sublingual glands [36,39,117,118]. Unilateral involvement is uncommon (around 10% of cases), leading to mass-forming fibrosing sialadenitis being misdiagnosed as tumor (e.g., Küttner’s tumors) [39,117,119]. CT images typically show a diffuse, homogeneous mass with regular borders or a “crazy paving” appearance in contrast enhancement [118,120], in contrast to neoplasms which generally have irregular margins with areas of necrosis or intratumoral cysts [118,120,121].

As in the case of Case 3, given the overlapping features with malignancy and other autoimmune diseases, often the diagnosis is performed after histopathological evaluation, which is crucial for the diagnosis of S-IgG4-RD [36,119,120]. Key histological findings include intralobular fibrosis, acinar atrophy, and lymphoplasmacytic infiltrate [38,39,118]. The diagnostic threshold for IgG4 positive plasma cells may vary from a value exceeding 10 cells/HPF in biopsy specimen and over 100 cells/HP in surgical samples, along with an IgG/IgG4 ratio greater than 0.4 [38,39,118].

## 4. Therapy

Some patients experience spontaneous regression and sustained remission from the disease without the need of therapy [14,122,123]. Particularly in patients with salivary gland or lymphnode involvement, watchful wait approach may be considered to avoid immunosuppressant’s side effects [122,123]. For the Case 3 patient, as there were no other signs of disease after the sialoadenectomy, we preferred to undergo a close follow-up and delay starting a treatment till eventual relapse.

IgG4-RD shows an excellent response to glucocorticoids [2,14]. The recommended induction regimen is prednisone (or equivalent) 0.6–1 mg/kg/day for at least 1 month, with response typically observed within the first 2–4 weeks [2,14,29,124]. For rapid remission, high-dose steroid intravenous pulses (125–1000 mg IV methylprednisolone) may be considered [29]. Although a higher steroid intensity approach is equally effective in inducing remission, it has been associated with a lower relapse rate [125]. Yoshifume and Umehara suggest that a medium-dose regimen (0.6–0.8 mg/kg/day) may be used as the starting dosage [124]. For the Case 2 patient, we preferred the pulses approach as her symptoms were worsening at the time the therapy started, and to load the patient with a lower glucocorticoid burden, a steroid-sparing therapy was therefore promptly started as outlined below.

Steroid response should be evaluated using clinical, radiological, biochemical, and, when necessary, pathological data as a whole [2,29]. In cases with established fibrosis, the response may be delayed or less evident [2,82]. Except for IgG4-RD of kidney, the absence of a response should prompt further and deeper investigations, including the option of repeating tissue histological evaluation by biopsies in suspected involved tissues, to exclude other possible diagnoses [2,29,59].

There is no consensus on steroid tapering; however, rapid tapering is associated with a higher risk of relapse, which usually happens between 3 and 16 months [29,124,126]. Recent studies advocate a prolonged tapering regimen, reducing the dose by approximately 10% (about 0.05 mg/kg/day) every two weeks to effectively prevent relapse [29,124].

Even after remission, the relapse rate remains high (40–90%) within the first three years from diagnosis, occurring either during tapering or after therapy withdrawal [29,30,124].

There is no clear evidence for a relapse marker. IgG4 levels may vary but have poor predictive value [126] and Responder Index > 9 seems to correlate with risk of relapse, along with high IgG4 levels and severe plasma cell infiltration, as described by Zongfei et al. [127]. Hence, we suggest a clinical follow-up period of 3–6 months for the first 2–3 years based on severity of disease and response to therapy. Relapses, which may occur in the same or different organs, generally still respond to the same steroid approach, requiring maintenance therapy [29].

Low-dose glucocorticoids have been proposed; however, their long-term use carries well-known major adverse effects, such as osteoporosis, diabetes mellitus, arterial hypertension, and increased risk for cardiovascular diseases [30,124]. Currently, there is no clear evidence supporting the efficacy on DMARD monotherapy as a steroid-sparing strategy. Azathioprine (2–2.5 mg/kg/day) and mycophenolate mofetil (1–2 g/day) are the most commonly used drugs, although evidence is largely based on retrospective studies and case reports [14].

Rituximab remains the most extensively studied drug, and apart from glucocorticoids, it is the only therapy capable of inducing and maintaining remission [14,124,126]. In a cohort study, Backhus et al. described that patients treated with rituximab, either 1 g given twice every 14 days or 375 mg/m^2^ given four times over the time period of four weeks, experienced an increase in relapse-free survival from 9 to 21 months and an improved response index, as compared to those patients using steroids as monotherapy [126]. Current guidelines suggest the use of rituximab in patients who are glucocorticoid-resistance or have contraindications, with consolidation dosing every six month to maintain the remission for up to two 2 years [14]. For the Case 1 patient, we considered the disease to be steroid-resistant and therefore we started immediately with rituximab. As long-term steroid therapy may have severe impact on patients’ health, we also decided to use rituximab as steroid-sparing therapy.

## 5. Conclusions

IgG4-RD is a rare autoimmune disease that, for its multifaceted and complex presentations, is often misdiagnosed as malignancy or as other clinical conditions, leading to a significant diagnostic delay and/or inappropriate treatment. The clinical presentation with plenty of atypical features, many of which remain incompletely defined and somewhat underrecognized, especially when they are not associated with more typical characteristics of IgG4-RD, usually further complicate the diagnostic process.

Serum IgG4 concentration shows limits as a diagnostic tool, with its low specificity and sensitivity, which may be increased by evaluation of IgG4/IgG ratio greater than 10% and IgG4/IgG1 ratio exceeding 21%.

Although extremely rare, GE-IgG4-RD should be considered in those cases where a clinically suspected IBD is not pathologically confirmed, particularly when gastroenteric wall thickening or strictures are present. A transient, even partial, improvement with short courses of steroids, followed by symptom relapses upon discontinuation, may provide further evidence raising suspicion for IgG4-related disease.

In instances where thoracic and abdomen effusions are identified as inflammatory without any other plausible cause, once cancer is excluded, SM-IgG4-RD or pleural involvement by IgG4-RD should be suspected.

O-IgG4-RD is often overlooked—unless severely symptomatic—but should be considered in the differential diagnostic path of patients with unexplained systemic symptoms and ocular involvement. When lacrimal glands are affected, they provide an accessible site for biopsy, thereby aiding in the confirmation of the diagnosis. Isolated meningeal involvement is uncommon but, if associated with other clinical features, can serve as supportive diagnostic evidence.

Both SC-IgG4-RD and S-IgG4-RD are frequently discovered fortuitously during evaluation for a suspicion of malignancy or other autoimmune disease. Similar to autoimmune pancreatitis, these manifestations should be included in the differential diagnosis to avoid unnecessary radical surgery. CT, MRI, and endoscopic ultrasonography can aid the clinician in differentiating SC-IgG4-RD from PSC and CCa.

In summary, IgG4-RD should be considered in patients with unexplained symptoms and lacking a definite diagnosis, particularly in those presenting mass-forming lesions or enlarged anatomical structures, and especially in subjects who responded to steroid therapy but experienced relapses upon therapy withdrawal.

We suggest that, in the appropriate clinical context, clinicians should assess serum or other biological fluid IgG4 levels and actively investigate for possible multi-organ involvement to support the diagnosis. Multidisciplinary collaboration with other specialists and the pathologists, who should be informed about patient’s clinical conditions, is crucial to effectively guide the diagnostic decisions and confirm the presence of IgG4-positive plasma cells on histological specimen, that otherwise would be overlooked (Figure 4). Furthermore, an experienced pathologist, if informed about the patient’s clinical feature, might suggest the most efficient diagnostic pathway and obtain a more precisely aimed biopsy.

## Figures and Tables

**Figure 1 diagnostics-15-02299-f001:**
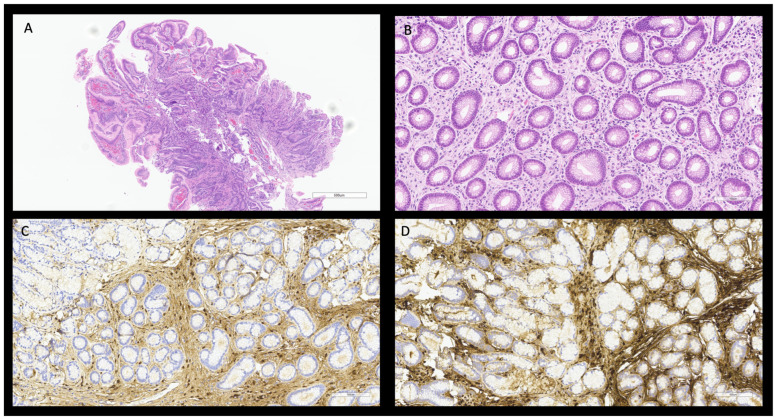
Histological features of gastric mucosae with foveolar hyperplasia H&E, 4× (**A**) and mild plasma cellular infiltrate H&E, 20× (**B**). The majority of IgG-positive plasma cells, immunostaining, 20× (**C**) appear positive for IgG4 20× (>10 plasma cells/HPF—ratio > 0.4), (**D**).

**Figure 2 diagnostics-15-02299-f002:**
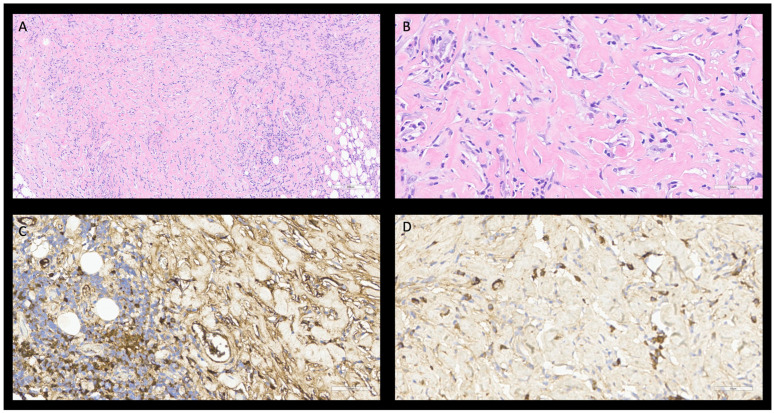
Histological features of IgG4-related sclerosing cholangitis typically show mild plasma cellular infiltrate H&E, 20× (**A**) with irregularity whorled pattern of fibrosis (storiform fibrosis), H&E, 20× (**B**). Majority of IgG4-positive plasma cells, immunostaining, 20× (**C**) appear positive for IgG4, 20× (>10 plasma cells/HPF—ratio > 0.4), (**D**).

**Figure 3 diagnostics-15-02299-f003:**
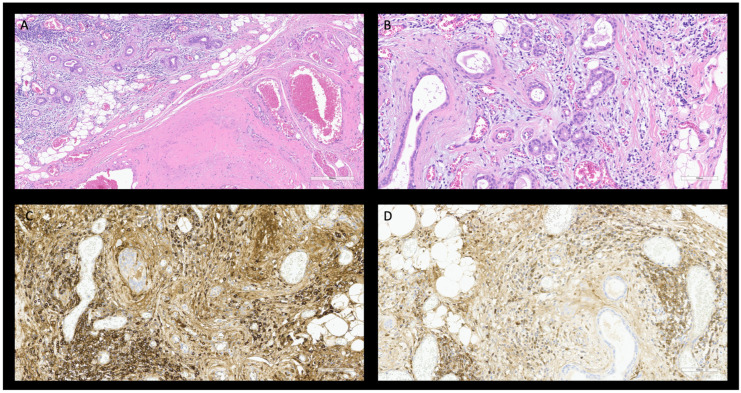
Histological features of IgG4-related sialadenitis show a mass-forming lesions with atrophic ducts, periductal fibrosis, with obliterated vein and mild plasma cellular infiltrate H&E, 20× (**A**,**B**). Majority of IgG-positive plasma cells, immunostaining, 20× (**C**) appear positive for IgG4 20× (≈95/105 plasma cells/HPF—ratio > 0.4), (**D**).

**Figure 4 diagnostics-15-02299-f004:**
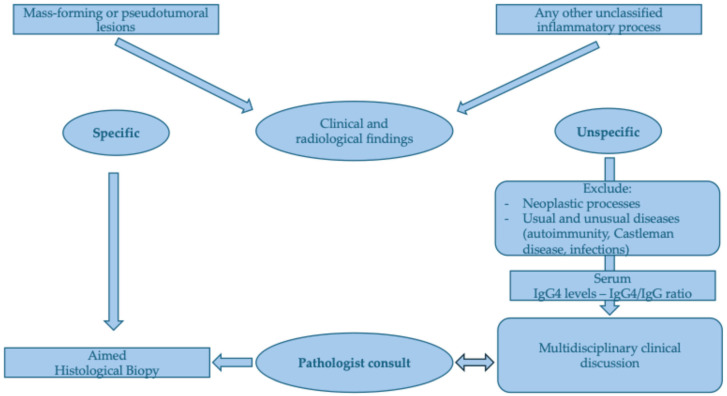
Practical monocenter approach for typical and atypical IgG4-RD manifestations. For specific organ involvement refer to Table 2.

**Table 1 diagnostics-15-02299-t001:** Case series summary.

	Case 1	Case 2	Case 3
**Clinical** **presentation**	Hyperemesis, diarrhea and stypsis, proptosis, symptoms relapse after steroid withdrawal.	Asthenia, loss of appetite, weight loss, jaundice.	Mild xerostomia, suspicious submandibular mass.
**Serology**	IgG4: 334 mg/dL, CRP 10–30 mg/dL. ANA negative TSH 1 UI/L.	IgG4 157 mg/dL, CRP 10–30 mg/dL. ANA negative. Bilirubin 7.85 mg/dL.	IgG4 24 mg/dL (after surgery). ANA negative.
**Imaging**	**Abdomen CT**: ascites, pleural effusion, thickened bowel with strictures. **Brain and Abdomen MRI**: ascites, bowel strictures, gastric wall thickening with signs of pachymeningitis **^18^FDG-PET/CT**: Gastric, colonic and bone marrow uptake.	**Abdomen CT**: mass-forming, infiltrative lesion surrounding the main bile duct (3.5 cm diameter), satellite lesions to gallbladder and right lung, highly suspicious for metastatic cholangiocarcinoma. **^18^FDG-PET/CT**: Not performed.	**US**: mass-forming expansive lesion of submandibular gland (2.5 cm diameter), satellite lymph nodes, suspicious for adenocarcinoma. **^18^FDG-PET/CT**: mild head–neck lymphnode uptake.
**Other tests**	**CSF**: mild protein increase. **EGDS**: signs of nonspecific chronic gastropathy. **Colonoscopy**: signs of nonspecific chronic colitis. **Explorative laparoscopy**: bowel wall and omental thickening, mesenterial nodule (3 cm diameter).	**ERCP and EUS**: Confirmed lesion of main bile duct. **Explorative laparoscopy**: firm, woody, infiltrative lesion at the hepatoduodenal ligament involving vascular structures.	**Surgery**: Sialoadenectomy.
**Pathology**	**Colon biopsy**: nonspecific chronic colitis. **Laparoscopy biopsies**: inflammatory peritoneal and omental fibrosclerotic nodules with no significant lymphoplasmacytic infiltrate; steroid therapy ongoing. **Gastric biopsy**: Chronic gastritis with fibrosis, IgG4 plasma cells > 10× HPF, IgG4/IgG > 0.4.	**EUS-FNB:** No malignant cell, presence of nonspecific fibroconnective tissue. **Intraoperative biopsy:** storiform fibrosis and phlebitis with IgG4 plasma cells > 10× HPF, IgG4/IgG > 0.4. Negative for cholangiocarcinoma.	**Submandibular gland histology**: storiform fibrosis and phlebitis with IgG4 plasma cells ≈ 95–105× HPF, IgG4/IgG > 0.4. Negative for adenocarcinoma.
**Therapy**	Steroid pulses and rituximab	Steroid pulses and rituximab	Adenectomy
**Outcome**	Symptoms solved; gastric biopsy fully restored	Mass reduced; symptoms solved	No other sign of disease

**Table 2 diagnostics-15-02299-t002:** Features of IgG4-RD by organ involvement.

Involved Site	Symptoms Related to the Involved Site	Possible Misdiagnosis	Serology	Imaging	Pathology	Other Commonly Performed Tests
**Lymphnode** [20,27,28,29]	Asymptomatic Single or multiple lymphadenopathy Local compression symptoms	Lymphoma Metastatic neoplasms Multicentric Castleman disease	ANA, RF negative RDSAbs negative IgG4 >135 mg/dL, usually >350 mg/dL IgG4/IgG ratio >10% IgG4/IgG1 ratio >24%	**US**: multiple lymphadenopathy. **CT**: lymphadenopathy (1–3 cm) with homogeneous contrast enhancement. **PET-CT**: ^18^FDG lymphadenopathy uptake.	**Histology:** Storiform fibrosis Obliterative phlebitis. Dense lymphoplasmacytic infiltrate. **Immunophenotype:** IgG4 plasm-cells > 50 cell/HPF IgG4/IgG ratio > 40%.	**Electrophoresis**: negative for monoclonal component. **BMB**: negative for lymphoproliferative disease.
**Pancreas** [14,20,29,30,31]	Diarrhea Weight loss Jaundice Diabetes	Pancreatic malignancy Other pancreatitis Lymphoma	ANA, RF negative RDSAbs negative Anti-SPINK1, antilactoferrin, anti-carbonic-anidrase II, anti-trypsinogen may be positive IgG4 > 135 mg/dL, usually >350 mg/dL IgG4/IgG ratio > 10% IgG4/IgG1 ratio > 24%	**CT**: pancreas enlargement with “sausage-like” shape, main duct stricture, no upstream dilation, edematous peripancreatic fat tissue and capsule-like rim. Delayed contrast enhancement. **MRI**: like CT with lower T1 SI with loss of pancreas lobulation, higher T2 SI, impeded water diffusion signal. Delayed contrast enhancement. **PET-CT**: patchy increase of ^18^FDG uptake.	**Histology:** Storiform fibrosis Obliterative phlebitis Dense lymphoplasmacytic infiltrate. **Immunophenotype:** **Biopsy**: IgG4 plasm-cells > 10 cell/HPF. IgG4/IgG ratio > 40%. **Surgical specimen**: IgG4 plasm-cell > 50 cell/HPF. IgG4/IgG ratio > 40%.	**US**: pancreas enlargement, with low echogenicity. **EUS**: diffuse or focal hypoechoic pancreas enlargement, hyperechoic strands and foci, parenchyma lobularity, hyperechoic wall of the main pancreatic duct, narrowing of the main pancreatic duct, duct-penetrating sign. **MRCP and ERCP:** Long main pancreas duct narrowing with no upstream dilation, skipped narrowed lesions, narrowing branches.
**Retroperitoneal** [20,29,31,32,33]	Asymptomatic Lower back pain Abdomen pain Lymphadenopathy	Retroperitoneal malignancy Lymphoma Multicentric Castleman disease Aortic vasculitis	ANA, RF negative RDSAbs negative IgG4 > 99 mg/dL, usually around 450 mg/dL IgG4/IgG ratio > 10% IgG4/IgG1 ratio > 24%	**CT**: retroperitoneal mass-forming lesions with homogeneous delayed phase contrast enhancement, usually in perivascular areas, hydroureteronephrosis, aortitis with aneurismal lesions or stenosing lesions. **MRI**: High T2 SI as in edema and inflammation. **PET-CT**: increase of 18FDG uptake in active disease areas.	**Histology:** Storiform fibrosis Obliterative phlebitis. Dense lymphoplasmacytic infiltrate. **Immunophenotype:** IgG4 plasm-cell > 30 cell/HPF. IgG4/IgG ratio > 40%.	
**Biliary ducts** [20,21,29,34,35]	Diarrhea Weight loss Jaundice	Cholangiocarcinoma Primitive sclerosing cholangitis	ANA, RF negative RDSAbs negative IgG4 > 135 mg/dL IgG4/IgG ratio > 10% IgG4/IgG1 ratio > 24%	**CT**: diffuse or long-segmental bile duct stricture, with funnel image and bile duct concentric thickening. Single-layered late phase contrast enhancement [35]. **MRI**: thicker bile ducts wall with contrast enhancement, without liver lesion. **PET-CT**: usually not performed.	**Histology:** Storiform fibrosis Obliterative phlebitis. Dense lymphoplasmacytic infiltrate. **Immunophenotype:** **Biopsy**: IgG4 plasm-cells > 10 cell/HPF. IgG4/IgG ratio > 40%. **Surgical specimen**: IgG4 plasm-cell > 50 cell/HPF. IgG4/IgG ratio > 40%.	Bile IgG4 levels 13 mg/dL. **EUS**: Regular homogeneous bile duct wall thickening can involve the cystic duct and the gallbladder. **MRCP and ERCP**: funnel-shaped, long-segmental stenosis.
**Salivary** [20,29,36,37,38,39]	Asymptomatic Monolateral or bilateral mass-forming lesions Xerostomia	Benign head–neck diseases Salivary neoplasms Sjögren Syndrome Lymphomas	ANA, RF negative RDSAbs negative IgG4 > 135 mg/dL IgG4/IgG ratio > 10% IgG4/IgG1 ratio > 24%	**CT**: bilateral asymmetric salivary gland swelling with regular borders and homogeneous or crazy-paving appearance at contrast enhanced imaging. Rarely suspicious infiltrating pattern.	**Histology:** Storiform fibrosis with acinar atrophy Obliterative phlebitis. Dense lymphoplasmacytic infiltrate. **Immunophenotype:** **Biopsy:** IgG4 positive plasm-cells > 10 cell/HPF. IgG4/IgG ratio > 40%. **Surgical specimen:** IgG4 positive plasm-cell > 100 cell/HPF. IgG4/IgG ratio > 40%.	**US**: homogeneous bilateral asymmetric salivary gland swelling.
**Lacrimal gland** [20,29,40,41,42]	Upper eyelid swelling Xeropthalmia Xerostomia (synchronous salivary involvement)	Sjögren Syndrome	ANA, RF negative RDSAbs negative IgG4 > 135 mg/dL IgG4/IgG ratio > 10% IgG4/IgG1 ratio > 24%	**CT**: enlargement of lacrimal gland, often symmetrical, usually homogeneous with contrast enhancement. Nodular lesions may be evident in the gland.	**Histology**: Storiform fibrosis with acinar atrophy Obliterative phlebitis. Dense lymphoplasmacytic infiltrate. **Immunophenotype**: IgG4 positive plasm-cells > 40–100 cell/HPF. IgG4/IgG ratio > 40%.	
**Retrobulbar** **involvement** [29,40,43,44,45,46]	Diplopia Proptosis Eyelid swelling Eye movement impairment	Hyperthyroidism Sjögren Syndrome Lymphoma	ANA, RF negative RDSAbs negative IgG4 > 135 mg/dL IgG4/IgG ratio > 10% IgG4/IgG1 ratio > 24%	**CT**: mass-forming, pseudotumoral lesions with infiltration of retrobulbar fat tissue, or infiltration of EOM. Infiltration of supraorbital or infraorbital nerves.	**Histology**: Storiform fibrosis Obliterative phlebitis. Lymphoplasmacytic infiltrate. **Immunophenotype:** Biopsy IgG4 positive plasm-cells > 10 cell/HPF. IgG4/IgG ratio > 40%.	
**Meninges** [20,29,44,47,48,49,50]	Headache Ataxia Seizures Cognitive impairment Intracranial hypertension Vascular compression Nerve compression	Meningioma Glial neoplasms Lymphoma Dementia	ANA, RF negative RDSAbs negative IgG4 > 135 mg/dL, usually >200 mg/dL IgG4/IgG ratio > 10% IgG4/IgG1 ratio > 24%	**CT**: linear thickening or mass-forming meningioma-like lesion of dura mater. **MRI**: dura mater thickening with diffuse T2-hypointensity and hyperintense spots, T1 contrast enhancement, meningioma-like mass-forming lesions.	**Histology**: Storiform fibrosis Obliterative phlebitis. Lymphoplasmacytic and mild eosinophilic infiltrate. **Immunophenotype:** IgG4 plasm-cells > 10 cell/HPF. IgG4/IgG ratio > 40%.	**CSF**: mild protein increase, lymphocytosis, intrathecal production of oligoclonal IgG4 bands during active phase.
**Chest** **involvement** [20,29,51,52]	Asymptomatic Chest pain Asthenia Dyspnea Fever Cough Hemoptysis Asthma	Interstitial lung diseases Sarcoidosis Malignancy Connective tissue diseases Rheumatoid arthritis Multicentric Castleman Disease	ANA < 1:160, RF negative RDSAbs negative IgG4 > 135 mg/dL IgG4/IgG ratio > 10% IgG4/IgG1 ratio > 24%	**Thorax Rx**: pleural effusion **CT**: pleural thickening, pleural effusion, bronchial wall thickening, interstitial lung disease (with either ground-glass, reticular nodular or peri bronchial pattern), mediastinal adenopathy. **PET-CT**: increased ^18^FDG uptake in other organs.	**Histology**: Storiform fibrosis (may be absent) Obliterative phlebitis (may be absent), arteritis Lymphoplasmacytic infiltrate. **Immunophenotype:** **Lung biopsy:** IgG4 plasm-cells > 20 cell/HPF. IgG4/IgG ratio > 40%. **Lung surgical specimen** IgG4 plasm-cell > 50 cell/HPF. IgG4/IgG ratio > 40%. **Pleural specimen:** IgG4 plasm-cell > 50 cell/HPF. IgG4/IgG ratio > 40%. **Mediastinal biopsy:** IgG4 plasm-cell > 30 cell/HPF. IgG4/IgG ratio > 40%.	**Pleural liquid**: slight increase in IgG4 and adenosine deaminase. **Pleural liquid cytology**: negative for malignant cells.
**Sclerosing** **mesenteritis** [29,53,54,55,56,57,58]	Asymptomatic Weight loss Diarrhea or stypsis Abdominal pain Ascites Bowel obstruction	Chronic infective disease Neoplastic diseases with or without carcinomatosis Lymphoma Multicentric Castleman disease Other edema causes	ANA, RF negative RDSAbs negative IgG4 > 135 mg/dL IgG4/IgG ratio > 10% IgG4/IgG1 ratio > 24%	**CT**: increased mesenterial fat tissue density, well-defined, non-infiltrating mass-forming lesions. **MRI**: masses with signs of fibrosis or edema in T2 weighted images. **PET-CT**: mild increased ^18^FDG uptake areas suspicious for malignancy.	**Histology:** Storiform fibrosis, fat necrosis Obliterative phlebitis. Dense lymphoplasmacytic infiltrate. **Immunophenotype:** IgG4 plasm-cells > 40–100 cell/HPF. IgG4/IgG ratio > 40%.	**US**: well-defined intrabdominal masses, no splenomegaly.
**Kidney** [9,20,29,31,59]	Proteinuria, sometimes with nephrotic syndrome Progressive kidney function loss Acute kidney injury or failure	Systemic lupus erythematosus Sjögren related nephropathy ANCA-vasculitis Kidney malignancy Other tubular interstitial nephritis causes (drugs, infection, myeloma) Multicentric Castleman’s disease	ANA positive 30% cases, RF negative Anti-PLA2R may be positive RDSAbs negative Low serum C3 and C4 Blood eosinophilia IgG4 > 135 mg/dL IgG4/IgG ratio > 10% IgG4/IgG1 ratio > 24%	**CT**: low-contrast enhanced low-density lesions. Lesions with mass-forming pattern, bilateral cortex nodulations, wedgy pattern or patchy pattens. Whole kidney enlargement. Renal pelvis thickening. **MRI**: mass-forming lesions, low-iso T1 signal intensity, T2 signal intensity inversely proportional to fibrosis, mild or none contrast enhancement. **PET-CT**: ^18^FDG uptake area in other organs.	**Histology:** Storiform fibrosis with tubular atrophy or acute interstitial nephritis associated with dense lymphoplasmacytic infiltrate. Membranous glomerulonephritis with IgG4 deposits. Rarely arteritis or phlebitis. **Immunophenotype:** **Biopsy** IgG4 positive plasm-cells > 10 cell/HPF. **Surgical specimen** IgG4 positive plasm-cell > 30 cell/HPF. IgG4/IgG ratio > 40% (sometimes can be missing).	**US**: ill-defined areas of low echogenicity, diffuse kidney swelling.
**Skin** [20,29,60]	Red or brown swelling nodule or plaques or masses, frequently at head neck region. Psoriasis like lesions	Kimura disease Angio-lymphoid hyperplasia with eosinophilia Eosinophilic angiocentric fibrosis Granuloma faciale Cutaneous plasmacytosis Multicentric Castleman disease	ANA, RF negative RDSAbs negative IgG4 > 135 mg/dL IgG4/IgG ratio > 10% IgG4/IgG1 ratio > 24%	**CT**: signs of other organ involvement. **PET-CT**: signs of other organ involvement.	**Histology** Storiform fibrosis Obliterative phlebitis. Dense lymphoplasmacytic and eosinophilic infiltrate. **Immunophenotype:** IgG4 plasm-cell > 200 cell/HPF. IgG4/IgG ratio > 40%.	
**Gastroenteric tract** [29,46,61,62]	Asymptomatic or abdominal pain Nausea and vomit Stypsis or diarrhea	IBDs or other colitis Malignancy	ANA, RF negative RDSAbs negative IgG4 > 135 mg/dL, usually 350 mg/dL IgG4/IgG ratio > 10% IgG4/IgG1 ratio > 24%	**CT and MRI**: mass-forming lesions, focal or diffuse gastroenteric wall thickening, sub-stenosis, gastric and bowel nodules. **PET-CT**: diffuse ^18^FDG uptake, corresponding to active disease sites.	**Histology** Storiform fibrosis patches. Obliterative phlebitis. Bottom heavy lymphoplasmacytic infiltration. **Immunophenotype:** IgG4 plasm-cells > 10–50 cell/HPF. IgG4/IgG ratio > 40–50%.	**Endoscopy** submucosal mass or chronic mucosal inflammation, sometimes with ulceration or polypoid lesions.
**Liver (non-cholangitis** **involvement)** [20,29,63,64,65]	Dysgeusia Abdomen distension Generalized edema Liver failure	Other causes of cirrhosis Autoimmune hepatitis	ANA may be present, RF negative ASMA may be positive RDSAbs negative GOT, GPT, GGT increased IgG4 > 135 mg/dL IgG4/IgG ratio > 10% IgG4/IgG1 ratio > 24%	**CT** hepatic mass-forming lesions, liver cirrhosis and portal hypertension signs. **MRI** liver cirrhosis, portal hypertension signs, no bile ducts abnormalities. **MCRP** no bile ducts abnormalities.	**Histology** Fibrosis, portal inflammation, interface hepatitis, confluent necrosis, rosette formation. Obliterative phlebitis. Dense lymphoplasmacytic infiltrate. **Immunophenotype:** **Biopsy** IgG4 plasm-cells > 10 cell/HPF. IgG4/IgG ratio > 40%. **Surgical specimen** IgG4 plasm-cell > 50 cell/HPF. IgG4/IgG ratio > 40%.	**ERCP** no bile ducts abnormalities.

Abbreviation: ASMA: Anti Smooth Muscle Antibodies; RDSAbs: rheumatic disease-specific antibodies; RF: Rheumatoid Factor; ^18^FDG: 18Fluorodeoxyglucose; SI: Signal Intensity; EOM: Extra Ocular Muscles; BMB: Bone Marrow Biopsy; CSF: Cerebrospinal fluid; Rx: radiography; MRCP: magnetic resonance cholangiopancreatography.

## Data Availability

The original contributions presented in this study are included in the article/Appendix A. Further inquiries can be directed to the corresponding authors.

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
