# Peer review of "Clinicopathological Pearls and Diagnostic Pitfalls in IgG4-Related Disease: Challenging Case Series and Literature Review"

_diagnostics, 2025, doi:10.3390/diagnostics15182299_

Round 1
Reviewer 1 Report
Comments and Suggestions for Authors
Clinicopathological Pearls and Diagnostic Pitfalls in IgG4-Related Disease: Challenging Case Series and Literature Review
The manuscript covers an interesting topic. However, some points need to be addressed
- The goal of the manuscript is stretched out over the background and introduction. It might be stated more clearly and concisely at the beginning, explaining why these three unusual situations were chosen and what new insights they offer.
- The case narratives are very thorough, however they can be hard to follow because they mix clinical presentation, workup, and pathology findings in the same lines. It would be easier to understand if each case had clear subheadings like "Clinical Presentation," "Investigations," "Histopathology," "Treatment," and "Outcome."
- The literature review is thorough, however it is largely made up of several sections that come after the instances. The manuscript may be more cohesive if it took a more integrated approach, such as going over important review ideas with each example to show how they are similar or different.
- The background could include more recent epidemiological data and new information on the underlying mechanisms of immunopathogenesis to help explain unusual symptoms.
- To further explain the importance of the case series, stress early on that IgG4-RD is often misdiagnosed, especially when it affects rare organs.
- The manuscript talks about how hard it is to diagnose, but it should go into more detail on the problems with the present criteria, especially when it comes to less common organ involvement.
- Table 1 is useful, but it could be better if it had more standardized units, clearer column titles, and a list of the most important diagnostic problems or mistakes for each instance.
- Figures that illustrate histology should be high resolution and have scale bars for comparison. Adding IHC staining quantification would make things clearer.
- It talks about IgG4-RD relapses, but it doesn't give any specific advice on how to keep treating them, how often to get imaging or serological tests, or how to deal with typical problems.
- Some organ-specific symptoms have been linked to different diagnostic thresholds or therapeutic responses. Talking about these differences would give a more balanced view.
- Some information on histology or lab results is repeated too much in multiple parts. To cut down on redundancy, combine them.
- Always explain what an abbreviation means the first time you use it. Some acronyms, like HPF and EGDS, show up late without any explanation.
- Stress the need of collaborative clinical-pathological work by giving examples or suggestions for how to do so.
Author Response
Thank you very much for taking the time to review this manuscript. Please find the detailed responses below and the corresponding revisions and corrections highlighted in the re-submitted files:
1 - The goal of the manuscript is stretched out over the background and introduction. It might be stated more clearly and concisely at the beginning, explaining why these three unusual situations were chosen and what new insights they offer
we stretched out the goal better of the manuscript in the introduction as suggested, also we stated more clearly how the clinical cases add up to knowledge in the field of rarer IgG4-RD features, in particular for patients with gastroenteric disease (you can find it highlighted with a comment referring to your point)
2- The case narratives are very thorough, however they can be hard to follow because they mix clinical presentation, workup, and pathology findings in the same lines. It would be easier to understand if each case had clear subheadings like "Clinical Presentation," "Investigations," "Histopathology," "Treatment," and "Outcome."
In the most complex case we added subheadings, as requsted, in the other two cases we thought it would split the case in smaller pieces that would impair the reading.
3 - The literature review is thorough, however it is largely made up of several sections that come after the instances. The manuscript may be more cohesive if it took a more integrated approach, such as going over important review ideas with each example to show how they are similar or different.
We added a comments in literature review referring to the various case where appropriated, highlighting the similarities and the differencies as requested. (you can find them highlighted with a comment referring to your point)
4 - The background could include more recent epidemiological data and new information on the underlying mechanisms of immunopathogenesis to help explain unusual symptoms.
the goal of this work was to underline the unusual clinical feature of the disease, not to explain its pathogenesis, as far as we know there is no publications relative to different pathogenetic mechanism that could explain the rarer involvement. Epidemiological updated data were added.
5 - To further explain the importance of the case series, stress early on that IgG4-RD is often misdiagnosed, especially when it affects rare organs.
We assessed earlier the point that IgG4-RD is often misdiagnosed, usually as cancer, earlier in the manuscript (you can find the revision in the text highlighted with a comment referring to your point)
6 - The manuscript talks about how hard it is to diagnose, but it should go into more detail on the problems with the present criteria, especially when it comes to less common organ involvement.
we added a comment on the present criteria, highlighting their limitations and the difficulties in diagnosis, especially when rarer organ involvement is present (you can find them in the text highlighted with a comment referring to your point)
7 - Table 1 is useful, but it could be better if it had more standardized units, clearer column titles, and a list of the most important diagnostic problems or mistakes for each instance.
table 1 is relative to the three clinical cases, it was set to summarize the main features of the cases and to highlight similarities and differences between them. We tried to better highlight the columns titles also in table 2, and modified those that where less clear. All measure unit are expressed in internation measure system units, some values are measureless (as ratios) or expressed as cell per High Power Field as stated in literature
8 - Figures that illustrate histology should be high resolution and have scale bars for comparison. Adding IHC staining quantification would make things clearer.
figures are high definition and each one of them is provided with a scale bar in the right bottom corner, IHC staining was set by the pathologist with visual quantification.
9 - It talks about IgG4-RD relapses, but it doesn't give any specific advice on how to keep treating them, how often to get imaging or serological tests, or how to deal with typical problems.
as far as we know, no criteria are available today to chose a timing for follow-up nor to change treatment. Clinical, laboratoristical and instrumental follow-up are decided on bases of clinical manifestation and response to first-line treatment (steroids and rituximab). Relapses are treated and usually respond to first-line treatment. Response index seems to be linked to relapse risk but only few evidence is present and we found no other clear indication on timing. We added a comment on this at the end of treatment part of the manuscript. (you can find it highlighted with a comment referring to your point)
10 - Some organ-specific symptoms have been linked to different diagnostic thresholds or therapeutic responses. Talking about these differences would give a more balanced view.
Different indication for diagnostic cut-of are therefore indicated by scientific literature and we listed them to underline the etherogenity of the disease and a better diagnostic definition related to different organ involvement. We added a comment to highlight the differences and their meaning (you can find it highlighted with a comment referring to your point)
11 - Some information on histology or lab results is repeated too much in multiple parts. To cut down on redundancy, combine them.
We wanted the article to be intended also for everyday use, indeed during the writing of the paper we tried to combine them, but found that it would lower the everyday availability of the informations.
12 - Always explain what an abbreviation means the first time you use it. Some acronyms, like HPF and EGDS, show up late without any explanation.
All the abbreviation are listed at the end of manuscript, as well on the bottom of tables or in the text (Example EGDS line 148, pag 5)
13 - Stress the need of collaborative clinical-pathological work by giving examples or suggestions for how to do so.
The importance of interaction between clinicians and pathologist is underlined by opportunity for the pathologist to know clinical features and organ involvement of the patient, enabling him to better decide the correct test and search for the disease. We remarked the needing for clinical information for the pathologist in the conclusion section of the manuscript (you can find it highlighted with a comment referring to your point)
Thank for your precious advices

Reviewer 2 Report
Comments and Suggestions for Authors
This manuscript presents a case series of three patients with unusual clinical presentations of IgG4-related disease (IgG4-RD) that did not fit into the current classification criteria. The authors aim to highlight some of the most overlooked manifestations of the disease, focusing on characteristics that may either mislead or guide clinicians toward accurate and timely diagnosis. Notably, the manuscript emphasizes the difficulties in diagnosing IgG4-RD due to its variable clinical phenotypes and the unreliability of laboratory markers, and offers insights into the therapeutic management of IgG4-RD, including the use of glucocorticoids and rituximab..
The manuscript is of clinical interest; however, some issues should be addressed.
-Differential Diagnosis of Liver Involvement: distinguishing IgG4-related sclerosing cholangitis (SC-IgG4-RD) from primary sclerosing cholangitis (PSC) is a critical issue. Expanding on this would be valuable, in particular, specific imaging features that can help differentiate the two (e.g., single-layered enhancement in SC-IgG4-RD vs. double-layered in CCa).- The importance of EUS and intraductal ultrasound (IDUS) for targeted biopsies.
-Role of Antinuclear Antibodies (ANA): The point about ANA patterns is excellent. The manuscript should emphasize that while ANA elevations can occur in IgG4-RD, certain patterns (especially "multiple nuclear dots and rim-like/membranous patterns") are more suggestive of PBC, a more frequent chronic cholestatic liver disease, and warrant further investigation for that condition, as previously demonstrated (DOI: 10.1586/erm.11.82) . It's important to state that other ANA patterns are less specific and can be found in other liver diseases (HCV, HBV, etc.).
Author Response
Thank you very much for taking the time to review this manuscript. Please find the detailed responses below and the corresponding revisions/corrections highlighted in the re-submitted files
1-Differential Diagnosis of Liver Involvement: distinguishing IgG4-related sclerosing cholangitis (SC-IgG4-RD) from primary sclerosing cholangitis (PSC) is a critical issue. Expanding on this would be valuable, in particular, specific imaging features that can help differentiate the two (e.g., single-layered enhancement in SC-IgG4-RD vs. double-layered in CCa).- The importance of EUS and intraductal ultrasound (IDUS) for targeted biopsies.
We expanded the section relative to the imaging differential diagnosis, either for CT, MRI and endoscopic ultrasound.
2- Role of Antinuclear Antibodies (ANA): The point about ANA patterns is excellent. The manuscript should emphasize that while ANA elevations can occur in IgG4-RD, certain patterns (especially "multiple nuclear dots and rim-like/membranous patterns") are more suggestive of PBC, a more frequent chronic cholestatic liver disease, and warrant further investigation for that condition, as previously demonstrated (DOI: 10.1586/erm.11.82) . It's important to state that other ANA patterns are less specific and can be found in other liver diseases (HCV, HBV, etc.).
We thank you for the interestin paper linked, we expanded the role of ANA testing in the differential diagnosis and their importance to suspect other diseases.
Thank you for your precious advices

Reviewer 3 Report
Comments and Suggestions for Authors
The manuscript presented three cases of IgG4 related disease (IgG4 RD), and a review for IgG4 RD. The review seemed to be sound and in details. However, due to most of IgG4 RD cases were not with specific or typical symptoms, the cases seemed to be not very interesting. The authors may consider revise the manuscript as a “ review article” instead of case report. The review article may include the three case reports.
Comments:
- the manuscript may be considered to be revised as review article.
- If the manuscript is kept as case reports. Suggest the authors may add more information for the case presentation. Case 1, UGI/LGI endoscopy picture, CT, and laparoscopic findings image. Case 2. CT, ERCP, EUS findings. Case 3. The ultrasonography finding.
- The cases use pulse therapy and then rituximab. The authors may provide their considerations.
- Despite of mention about clinical alert, the conclusion section dose not clearly provide hints for clinical suspicious of IgG4 RD.
5.the abbreviation of each tables should be moved to the bottom of tables.
Author Response
Thank you very much for taking the time to review this manuscript. Please find the detailed responses below and the corresponding revisions/corrections highlighted in the re-submitted files
1 - The manuscript may be considered to be revised as review article.
We would be honored to have the manuscript revised as literature review, in fact, initially we submitted it as review, but the editor suggested us to change it into case report. If you may help us in that sense and push the editor into reclassify it as review we would be glad to you.
2 -If the manuscript is kept as case reports. Suggest the authors may add more information for the case presentation. Case 1, UGI/LGI endoscopy picture, CT, and laparoscopic findings image. Case 2. CT, ERCP, EUS findings. Case 3. The ultrasonography finding.
With the exception of case 1 laparoscopic images, pictures of the cases are all available, yet we thought adding all of them to the text would might impair the readability of the article. We will add them as supplemental materials, we already have sent a mail to MDPI to do so, thank for your consideration
3 - The cases use pulse therapy and then rituximab. The authors may provide their considerations.
Added considerations about therapy with rituximab and pulse steroids in the therapy sections, we used those approach both to spare steroid burden and as rescue therapy in the gastroenteric IgG4-RD who had only a partial response to steroid
4- Despite of mention about clinical alert, the conclusion section dose not clearly provide hints for clinical suspicious of IgG4 RD.
we expanded conclusion section to better include hints for clinical suspicion for IgG4RD
5 - the abbreviation of each tables should be moved to the bottom of tables.
we moved abbreviations to the bottom of tables
Thank you for your precious advices

Reviewer 4 Report
Comments and Suggestions for Authors
I would like to congratulate the author on this series of three challenging cases of IgG4-related disease. The literature review is well-structured and offers a clear explanation of the various cases, along with comprehensive details on how different organs are involved.
Although I would like to make some comments:
1.- In the abstract, I recommend providing more specific details about the three cases.
2.- In the introduction, it would be beneficial for the reader to provide some data regarding the prevalence of the different types mentioned in lines 65-70 (page 2).
3.- It would enhance the case report to better articulate why some challenging and unusual presentations of IgG4-RD cases, like those presented, could contribute to the existing literature (page 3, lines 98-102)-
4.- Within Table 1, I suggest adding the value of IgG plasma cells in the submandibular gland histology.
5.- I recommend providing explanations for cases 1 and 3 regarding why corticosteroids were not started, and for case 1, to initiate Rituximab directly.
6.- For case 3, since the PET-CT scan indicated hypermetabolism in the right colon, was a colonoscopy performed to rule out any colonic involvement?
Author Response
Thank you very much for taking the time to review this manuscript. Please find the detailed responses below and the corresponding revisions/corrections highlighted in the re-submitted files
1.- In the abstract, I recommend providing more specific details about the three cases.
modified the abstract to fit more details of the three cases, as well to fit the 250 words.
2.- In the introduction, it would be beneficial for the reader to provide some data regarding the prevalence of the different types mentioned in lines 65-70 (page 2).
Added prevalence and incidence of the overall disease, also in text added the prevalence as stated by Wallace et al. in 2019
3.- It would enhance the case report to better articulate why some challenging and unusual presentations of IgG4-RD cases, like those presented, could contribute to the existing literature (page 3, lines 98-102)-
Added a comment in the text to better explain the needing of further evidence about unusual presentations of IgG4 RD, as well as a comment about the current classification criteria not being adequate for diagnostic purposes
4.- Within Table 1, I suggest adding the value of IgG plasma cells in the submandibular gland histology.
added IgG4 plasma-cells value in table
5.- I recommend providing explanations for cases 1 and 3 regarding why corticosteroids were not started, and for case 1, to initiate Rituximab directly.
Added to text both for cases: in case 1 we considered the already administered corticosteroids as a faiure therapy, for case 3 there was only salivary gland involvement, disease went in spontaneus remission after the surgery and the patient is still in follow-up, if ever a new symptom would appear we will start corticosteroids
6.- For case 3, since the PET-CT scan indicated hypermetabolism in the right colon, was a colonoscopy performed to rule out any colonic involvement?
Colonic involvement was excluded via colonoscopy, whose result we added in text, thank you for the point.
Thank you for all the precious advices
Round 2
Reviewer 1 Report
Comments and Suggestions for Authors
The manuscript improved
Author Response
Thank you very much for your time
Reviewer 3 Report
Comments and Suggestions for Authors
The authors presented three case series and review of IgG4 related disease. The authors provide an extensive revised manuscript. However, this type of rely to reviewer comments seemed to be not a typical type of response.
- Despite of a type of review article was not accept by the authors, as a three case series seemed to be acceptable.
- The authors have provided the image of the cases as supplementary data.
- The conclusion appears somewhat redundant. Providing an algorithm outlining the diagnostic approach to typical and atypical IgG4-related disease would offer a clearer and more easily understandable concept for readers.
Author Response
Dear revisor, please find the detailed responses below and the corresponding revisions changes in the re-submitted files:
1. Despite of a type of review article was not accept by the authors, as a three case series seemed to be acceptable.
We modified the manuscript as review.
3. The conclusion appears somewhat redundant. Providing an algorithm outlining the diagnostic approach to typical and atypical IgG4-related disease would offer a clearer and more easily understandable concept for readers.
We built a practical flow-chart approach alogorithm referring to table II for further details. You can find it at conclusion section.
Thank for your suggestions,
sincerely
Sokol Sina
Reviewer 4 Report
Comments and Suggestions for Authors
The authors have responded appropriately to my comments and have clearly improved the manuscript's presentation, so I have no further comments.
Author Response
Thank you very much for your time